# Exploring Antibacterial Usage and Pathogen Surveillance over Five Years in a Tertiary Referral Teaching Hospital Adult General Intensive Care Unit (ICU)

**DOI:** 10.3390/pathogens13110961

**Published:** 2024-11-05

**Authors:** David Young, Cathrine A. McKenzie, Sanjay Gupta, David Sparkes, Ryan Beecham, David Browning, Ahilanandan Dushianthan, Kordo Saeed

**Affiliations:** 1Pharmacy Department, University Hospital Southampton NHS Foundation Trust, Southampton SO16 6YD, UK; 2Cancer Sciences, Faculty of Medicine, University of Southampton, Southampton SO17 1BJ, UK; 3National Institute of Health and Social Care Research (Biomedical Research Centre Southampton Perioperative and Critical Care Theme), Faculty of Medicine, University of Southampton, Southampton SO17 1BJ, UKahilanadan.dushianthan@uhs.nhs.uk (A.D.); kordo.saeed@uhs.nhs.uk (K.S.); 4General Intensive Care Unit, University Hospital Southampton NHS Foundation Trust, Southampton SO16 6YD, UK; sanjay.gupta@uhs.nhs.uk (S.G.);; 5Microbiology Department, University Hospital Southampton NHS Foundation Trust, Southampton SO16 6YD, UK

**Keywords:** antimicrobial stewardship, antibacterial consumption, pathogen surveillance, COVID-19 pandemic

## Abstract

Antimicrobial resistance is a globally recognised health emergency. Intensive care is an area with significant antimicrobial consumption, particularly increased utilisation of broad-spectrum antibacterials, making stewardship programmes essential. We aimed to explore antibacterial consumption, partnered with pathogen surveillance, over a five-year period (2018 to 2023) in a tertiary referral adult general intensive care unit (ICU). The mean number of admissions was 1645 per annum. A comparison between the ICU populations admitted before and after the COVID-19 pandemic peak (2020/21) identified several notable differences with increased average daily unit bed occupancy (21.6 vs. 25.2, respectively) and a higher proportion of admissions with sepsis (28.4% vs. 32.5%, respectively) in the post-pandemic period. Over the entire five years, the overall proportion of antibacterial use by the WHO AWaRe classification was 42.6% access, 54.7% watch and 2.6% reserve. One hundred and forty-seven positive blood culture isolates were reported, with the most concerning antibacterial resistance identified in 7.5% (9 *Escherichia coli* and 2 *Klebsiella pneumoniae* isolates). The COVID-19 pandemic peak year was associated with increased ICU bed occupancy, as well as a greater number of positive blood cultures but lower antibacterial consumption. Despite an increasingly complex workload, a large proportion of overall antibacterial consumption remained within the access category. However, the mortality rate and the incidence of most concerning antimicrobial resistance with respect to pathogens remained satisfyingly consistent, suggesting the positive consequences of real-world antibiotic stewardship in an intensive care setting.

## 1. Introduction

Antimicrobial resistance (AMR) is recognised as one of the most pressing threats to humanity. It has been estimated that drug resistance in infections was directly responsible for 1.27 million (95% uncertainty interval (UI) 0.91–1.71) deaths globally in 2019 with 4.95 million (95% UI 3.62–6.57) deaths associated with AMR, making this a leading cause of death [1]. The highest impact is seen in low-resource settings, particularly sub-Saharan Africa and South Asia [1]. Judicious use of antimicrobials is essential to preserve the effectiveness of these medicines. Antimicrobial stewardship programmes (coordinated interventions designed to optimise the use of antimicrobial agents) are effective in improving patient outcomes and minimising adverse events [2]. Suggested antimicrobial stewardship interventions include education of clinicians, patients and the public, guidelines for the management of common infections and restricting access to specific antimicrobials [2].

The World Health Organisation (WHO) AWaRe classification of antibacterials was developed to support stewardship efforts. This stratifies antibacterials into three different groups (access, watch and reserve) based on their clinical importance and propensity for the development of antimicrobial resistance. “Access” antibacterials typically have a narrow spectrum of activity, have a lower risk of resistance selection and are suitable for empirical use, whereas the “reserve” antibacterials are expected to be used only in severe infections when essential according to sensitivity results [3,4]. Furthermore, use of internationally recognised standardisation for grouping antibacterial consumption facilitates accurate comparison. Defined daily doses (DDDs) are an internationally recognised and widely used method of assessing antimicrobial consumption. The DDD is defined as the assumed average maintenance dose per day for a drug used for its main indication in adults [5,6]. While useful for summary analysis, recognised limitations of utilising a standardised daily dose include failing to account for administered dose, the impact of patient-specific factors on posology and complex dosing regimens. The impact of the COVID-19 pandemic on antibacterial consumption, drug resistance and length of stay has not been fully described. A 2023 meta-analysis of outpatients and hospitalised patients with COVID-19 infection estimated the prevalence of bacterial co-infection and secondary bacterial infection to be 5.3% and 18.4%, respectively [7]. Among patients with bacterial infections, the proportion of isolates resistant to antimicrobials was 37.5%, highlighting that AMR is highly prevalent in patients with COVID-19 infection and worthy of surveillance through systematic collection and analysis of local pathogen data. The aim of our project was to explore antibacterial consumption, partnered with pathogen surveillance, over a five-year period in a tertiary referral adult general intensive care unit (ICU).

## 2. Materials and Methods

### 2.1. Registration

The service evaluation, performed at University Hospital Southampton NHS Foundation Trust, was registered on the local governance system (project number SEV/0679) and was approved by the critical care institutional review body (28 March 2024), in accordance with the UK Policy Framework for Health and Social Care Research. The evaluation was conducted in accordance with the Declaration of Helsinki. As anonymised retrospective observational data were the focus of this project, an ethics waiver was approved by the local critical care governance lead. An expert group was convened to provide oversight.

### 2.2. Patient Characteristics

Patient characteristics and unit activity data (including intensive care length of stay, age, gender, admission specialty, acute physiological and chronic health evaluation (APACHE II), and unit bed occupancy) were extracted from the unit’s Intensive Care National Audit and Research Centre (ICNARC) case mix programme report and ICU daily summaries spanning April 2018 to March 2023 [8]. Infection-specific demographics were selected from internationally defined terms which describe infections commonly seen in critical care. These were high-risk admissions from a ward, sepsis and septic shock on unit admission and admissions with a positive SARS-CoV-2 polymerase chain reaction (PCR).

### 2.3. Assessment of Antibacterial Consumption

For the purposes of this project, the expert group collectively decided to focus solely on antibacterial use, as a brief review of the aggregated data suggested that antibacterial use far exceeded both antifungal and antiviral use. Antibacterial usage was derived from the clinical information system (MetaVision, iMDsoft, Tel Aviv, Israel) by aggregating the total doses administered to patients recorded as admitted to the general intensive care unit over the five-year period. Erythromycin was omitted from the analysis, as it was predominantly prescribed locally (on a restricted basis) for its prokinetic properties [9]. The AWaRe classification and DDDs were applied to standardise and interpret the data which was indexed for ICU bed occupancy. In cases where a DDD was not available on the reference website, the expert group gave consensus about the value to be utilised. In order to facilitate precision in antibacterial prescribing, a specific mandatory data collection form (that included indication, infection site, course duration and review date) was completed at the time of initiation. Antibacterial durations were determined as the number of consecutive days that each antibacterial was administered.

### 2.4. Pathogen Surveillance

For pathogen surveillance, we obtained positive ICU blood culture details and accompanying antibacterial susceptibilities retrospectively for all admitted patients during the study period from the electronic microbiology information system (PathManager, CliniSys Solutions Ltd., Chertsey, UK). ICU-detected bacteraemia data were extracted from clinical microbiology databases, including patient samples where there was a high clinical suspicion of infection, including all ESBL/AmpC-producing bacteria (*Serratia* spp., *Morganella* spp., *Citrobacter* spp. and *Enterobacter* spp.), *Escherichia coli*, *Klebsiella pneumoniae*, *Staphylococcus aureus* and *Streptococcus pneumoniae*.

Blood samples were collected aseptically in standardised automated systems (Bact/Alert 70^®^ 3D blood culture system, bioMérieux, Marcy-l’Étoile, France). Bacterial duplicates, defined as the same pathogen with the same resistance profile isolated from the same patient and the same site of infection, were excluded within 14 days. Bacterial identification was performed using MALDI ToF analysis (Biotyper, Bruker AXS GmbH, Karlsruhe, Germany). Dependent on the organism identification, the relevant antimicrobial sensitivity testing was performed using EUCAST methodology [10].

### 2.5. Individual Roles of the Expert Team Members

The expert team comprised consultant intensivists, clinical pharmacists, a consultant microbiologist and an infection control nurse. Antibacterial data were collated and analysed by pharmacists. Pathogen surveillance data were collated and analysed by the microbiology staff. Consultant intensivists and a critical care data analyst provided patient characteristics, outcome and unit activity data.

### 2.6. Statistical Methods

Quantitative data were presented as means with standard deviations (SDs) or as medians with interquartile ranges (IQRs) depending on the normality of the underlying distribution. Discrete data were presented as numbers and percentages. To investigate the relationship between variables, we conducted correlation analyses, reporting Pearson’s correlation coefficient (r), a significance value of *p* < 0.05 and a 95% confidence interval for the correlation coefficient. All statistical analyses were conducted using R version 4.3.2 (R Core Team, Vienna, Austria).

## 3. Results

### 3.1. Patient Characteristics and Unit Activity

Over the five years from 2018 to 2023, a total of 8233 admissions to ICU were recorded. Despite the COVID-19 pandemic, the annual admission count remained consistent, as routine work (such as elective surgery) was put on hold, especially during the first surge of the COVID-19 pandemic. Unit expansion meant an increase in ICU beds from 25 to 31 in April 2022. Patient characteristics and unit activity data are reported in Table 1.

Some notable differences exist between patient populations encountered before and after the COVID-19 pandemic: the proportion of admissions with sepsis increased from a mean of 28.4% (two years before the COVID-19 peak) to 32.5% (two years following COVID-19 peak). In addition, the average annual number of patients admitted primarily for a clinical haematology indication increased from 29 to 40.5%

During the peak of the COVID-19 pandemic (2020/21), mean maximum daily unit bed occupancy increased from 21.67 ± 2.2 to 29.2 ± 12.9 patients with a maximum bed occupancy of 67 patients (22 January 2021; Figure 1). During this period, both the proportion of admissions with sepsis and the frequency of unit-acquired blood infections peaked. A shift in admission specialty was noted, with less haematology and more upper gastrointestinal surgery.

### 3.2. Antibacterial Consumption

DDDs were available for all recorded antibacterials on the WHO website [6] with the exception of the following four antibacterials, where estimated DDDs (decided by the expert group) were used in the analysis:Co-trimoxazole (parenteral and oral): 1.92 g;Colistimethate (parenteral): 9 megaunits;Metronidazole (oral): 1.2 g;Sodium fusidate (oral): 1.5 g.

A grand total of 126,950 doses of antibacterial medications was recorded over the five-year period in 6053 unique patients. The overall proportion of antimicrobial use by AWaRe classification was 42.6% access, 54.7% watch and 2.6% reserve (Figure 2).

The antibacterials with the highest consumption in each group were metronidazole (access), piperacillin/tazobactam (watch) and linezolid (reserve) (Table 2). Considering each year individually, 2020/21 (which corresponded with the worldwide COVID-19 pandemic) had the highest proportion of access medicine, as well as the lowest proportion of both watch and reserve antibacterial use.

Only three reserve antibacterials had significant usage (with indications extracted from the mandatory data collection form); the highest was linezolid (most commonly for intra-abdominal sepsis), daptomycin (endocarditis) and tigecycline (intra-abdominal sepsis).

#### Antibacterial Indications and Length of Course

The most common indications for treatment courses were lower respiratory tract infection, intra-abdominal sepsis, line infection, urinary tract infections and septic shock. Considering all antibacterial courses, the median [IQR] course length was 2 [1, 5] days. Over the entire five-year period, the median course length for piperacillin with tazobactam was 4 [2, 6] days with no increase from year one (4 [2, 6] days) to year five (4 [2, 6] days). For metronidazole, the overall median course length was 2 [1, 4] days with no increase from year one (2 [2, 4] days) to year five (2 [1, 4] days). For co-amoxiclav, the overall median course length was 3 [2, 4] days with no increase from year one (3 [1, 4] days) to year five (3 [2, 5] days). For cefuroxime, the overall median course length was 2 [1, 3] days with no increase from year one (2 [1, 3] days) to year five (2 [1, 3] days). Overall median course length for meropenem was 4 [2, 7] days with no increase from year one (4 [3, 7] days) to year five (5 [3, 7] days).

### 3.3. Microbiological Surveillance

A total of 9197 blood cultures were analysed over this period with 244 isolates. Over the five-year period, the annual number of blood cultures analysed increased by 77% (from 1186 in 2018/19 to 2108 in 2022/23). Deduplicated positive blood culture isolates (n = 147) are summarised in Figure 3. The most concerning antibacterial resistance pattern was identified in 7.5% (11/147) of deduplicated positive isolates (nine *Escherichia coli* isolates with resistance to third generation cephalosporins; two *Klebsiella pneumoniae* isolates with resistance to aminoglycosides, fluoroquinolones and third generation cephalosporins). No carbapenem resistance in *Klebsiella pneumoniae* isolates was identified.

### 3.4. Correlation Analyses

Correlation analyses were performed to examine relationships between patient characteristics, unit activity metrics and summarised pathogen surveillance data (Figure 4). The associations between individual blood culture isolates and the proportion of ICU patients with sepsis on admission is reported in Figure 5. *Streptococcus pneumoniae* was excluded from the analysis, as only a single isolate was identified during the reported period [11].

## 4. Discussion

This retrospective analysis describes the antibacterial use and all isolated bacterial pathogens from a large tertiary referral major teaching hospital ICU over a five-year period that included the COVID-19 pandemic. We have assessed quality of antibacterial usage utilising the WHO AWaRe classification, quantified by internationally accepted DDDs and via median course durations. The expert team were satisfied with overall usage stratified into AWaRe classification with exceptionally low usage of reserve agents (2.6%). In terms of assessing quantity, the overall antibacterial duration was deemed acceptable with a median overall course duration of 2 [1, 5] days. However, more focus could be given to minimising variation in antibacterial durations, especially meropenem, including the introduction of automatic stop dates on the electronic prescribing system.

Our five-year study period included the COVID-19 pandemic, which presented multiple challenges to health systems with a significant impact on critical care services, particularly in the early stages before vaccination when severe disease was relatively common. Prior to this, the proportion of commissioned beds occupied by a patient was approximately 90%. During the peak of the pandemic, the number of available beds on the ICU increased to 76; of these, 55 were exclusively for treating patients with COVID-19 infection. The consequent decrease in staff-to-patient ratio, expansion of the unit to external areas of the hospital and sharing of bed spaces may have impacted multiple factors, including antimicrobial resistance. Global antibacterial use (extrapolated from sales data) has been shown to be positively associated with COVID-19 cases [12]. It has been estimated that 75% of patients with COVID-19 infection received antibacterials, although secondary bacterial infection has been shown to be relatively rare (<10%) [12].

Some important differences were noted in the ICU populations before and after the pandemic peak year. By adjusting our antibacterial consumption data for bed occupancy, we accounted for the increase in average daily bed occupancy and length of stay. The increase in the proportion of patients admitted with sepsis and complex diagnoses undoubtedly contributes to increased antibacterial consumption. Conversely, the COVID-19 pandemic peak year was associated with increased ICU bed occupancy, as well as a greater number of positive blood cultures but lower antibacterial consumption. Additionally, the annual number of blood cultures analysed increased by 77% over the five-year period. We speculate that potential reasons for this increase include delayed presentation due to the COVID-19 pandemic and an increased awareness of sepsis. It may also be a consequence of more complex surgeries (exacerbated by delayed presentation).

Correlation analysis demonstrated a strong positive correlation between mean bed occupancy, sepsis on ICU admission and COVID-19 as reason for ICU admission. Higher ICU bed occupancy was associated with a strong correlation with patients admitted with sepsis and incidence of bloodstream infections during ICU admission. Interestingly, there was not a significant correlation between any of these variables and either mortality or ICU length of stay. None of the described organisms demonstrated a significant correlation with sepsis at ICU admission.

The incidence of the most concerning antimicrobial resistance with respect to pathogens remained consistent during the study period, and none of the following monitored resistance patterns were identified:Methicillin-resistant *Staphylococcus aureus;*Carbapenemase-producing *Enterobacterales.*

This contrasts with recent studies that found the greatest increase in resistant *Klebsiella* spp. and *Pseudomonas* against quinolones following COVID-19 pandemic in the ICU [13], although notably, our pathogen surveillance data were limited to ICU-detected bacteraemia. Unlike units from Romania and the Indian subcontinent, we did not experience a significant increase in resistance to carbapenems [14]. A study in Brazil found increasing resistance of *Klebsiella* and *E. coli* in the context of COVID-19 with excessive antimicrobial consumption in the ICU, although the peak anti-infective consumption (February to May 2020) was not associated with the peak of microbiological isolates (January to April 2019) indicating untargeted use of essentially ineffective antimicrobials against the resistant isolates detected [15]. While the differences between these studies and our cohort may be explained by the nature of the patients cared for in our unit, we believe that the robust local multidisciplinary team approach employed, particularly with respect to antimicrobial stewardship, is germane. It is recognised that overprescribing of antibacterials, especially those used in the treatment of secondary bacterial infections in patients with COVID-19, can contribute to the selection of antibacterial-resistant bacteria [16,17,18]. In our unit, despite an increasingly complex workload, a large proportion of overall antibacterial consumption remained within the access category (according to the AWaRe classification), with the overall proportion of antibacterial use by AWaRe classification being 42.6% access, 54.7% watch and 2.6% reserve.

Additionally, the overall mortality rate remained consistent in our unit and the ratio of observed risk-adjusted mortality rate for all patients (based on the ICNARC model) was consistently below the nationally standardised expected values. This is possibly due to the strong multidisciplinary approach and surge planning during the COVID-19 pandemic. While the reason for the low mortality is multifactorial, there is evidence suggesting that overuse of antibiotics during COVID-19 in an attempt to reduce COVID-19 mortality in the short term may have contributed to long-term mortality from AMR [19]. These findings underscore the ongoing value of daily collaboration between clinical pharmacists, consultant microbiologists and ICU medical staff at the ward round, allowing for early escalation/de-escalation strategies and resulting in a robust antimicrobial stewardship program that promotes the careful use of these critical medicines.

The consultant intensivist is the overall leader of ICU patients’ care, responsible for overall decisions, rapid identification of sepsis and septic shock, identifying clinical scenarios where source control is paramount (e.g., necrotising fasciitis) and making the ultimate decision on antimicrobial therapy. The consultant microbiologist acts as the infectious disease specialist, bridging the gap between complex pathology findings and patient care decisions with a focus on diagnostics and antibacterial selection. The clinical pharmacist (consultant and senior colleagues) has responsibility for antibacterial optimisation in multiple organ failure and managing the risk/benefit balance of therapy (e.g., penicillin allergy de-labelling [20]), as well as reminding the consultant intensivists of antibacterial course duration. In the UK, three-quarters of pharmacists working in critical care prescribe antibacterial therapy in collaboration with and under the guidance of the consultant intensivist and microbiologists [21]. The infection prevention link nurse has oversight of infection prevention and control practices across the ICU and interfaces between these departments and the wider hospital infection prevention team. These actions are an essential component of a developing, coordinated global “OneHealth” approach to tackling AMR that includes managing the use of these medicines in veterinary medicine and agriculture, as well as minimising environmental factors that contribute to AMR transmission [22,23]. Our findings provide evidence supporting the value of a cohesive, multidisciplinary approach to combatting the threat of AMR. Looking forward, our future work will include guideline adherence for specific indications and medicine optimisation.

We recognise some limitations of this retrospective analysis. Data were collected from a single centre, and pathogen surveillance was limited to ICU-detected bacteraemia. Antibacterial dose administration data were limited to those associated with the ICU and drug consumption data were limited to antibacterials with no distinction made between prophylactic use and treatment courses. DDDs for four antibacterials were not found on the reference site, and so values were decided locally.

In conclusion, despite these limitations and an increasingly complex workload, a large proportion (42.6%) of overall antibacterial consumption remained within the access category. However, the incidence of the most concerning antimicrobial resistance (identified in only 11 blood cultures) with respect to pathogens and mortality rate remained satisfyingly consistent (approximately 84% of what was predicted). These results suggest the positive consequences of real-world antimicrobial stewardship and multidisciplinary work (pharmacists, microbiologists and intensivists) in an intensive care setting.

## Figures and Tables

**Figure 1 pathogens-13-00961-f001:**
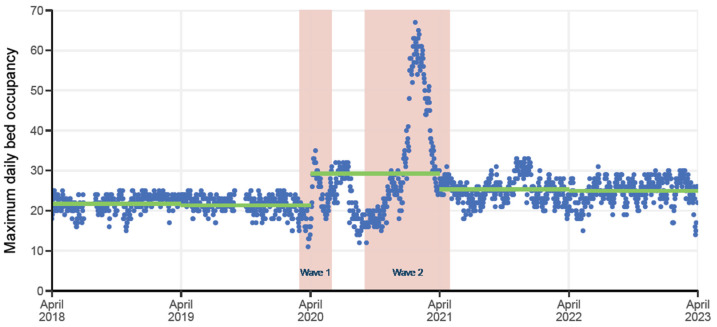
Maximum daily bed occupancy for the ICU with COVID-19 pandemic waves (pink) and annual mean maximum bed occupancy (green).

**Figure 2 pathogens-13-00961-f002:**
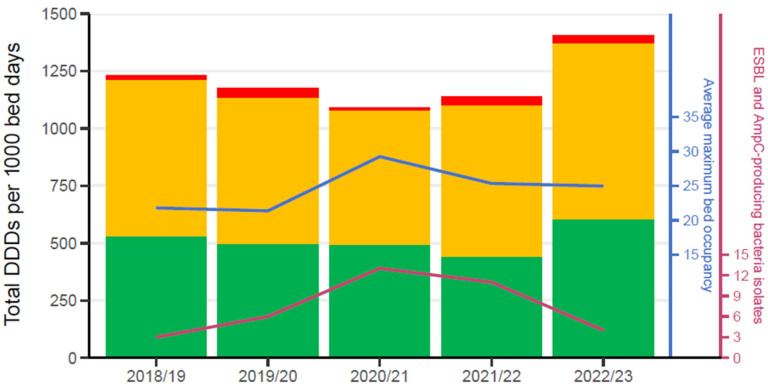
Annual (April to March-inclusive) antibacterial usage categorised according to the WHO AWaRe classification (access: green; watch: yellow; reserve: red).

**Figure 3 pathogens-13-00961-f003:**
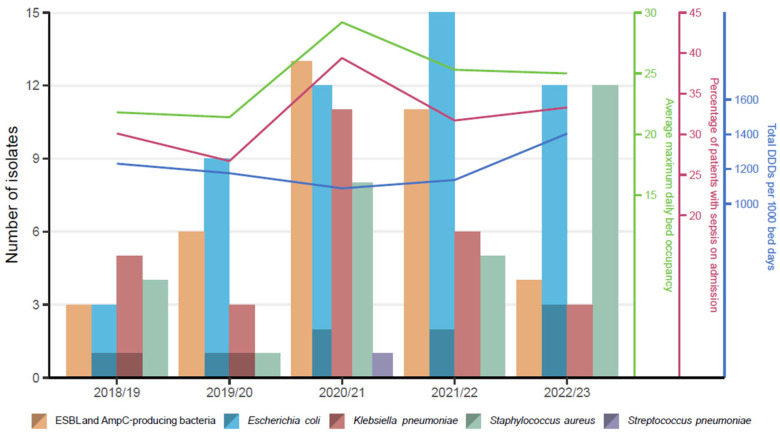
Positive blood culture isolates (isolates with the most concerning antibacterial resistance are shown in darker shade) by year (April to March-inclusive).

**Figure 4 pathogens-13-00961-f004:**
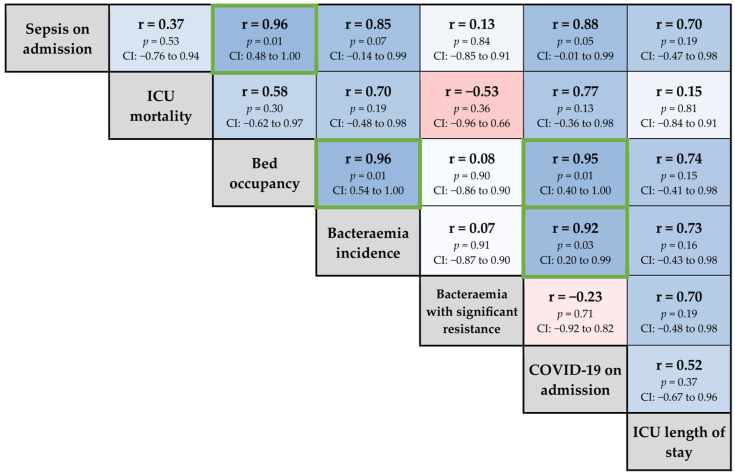
Correlation heatmap displaying pairwise Pearson correlation coefficients between patient characteristics, unit activity and pathogen surveillance. Colour intensity represents correlation strength and direction (darker shades correspond to stronger correlations, blue indicating positive correlations and red indicating negative correlations). Green borders highlight statistically significant correlations (*p* < 0.05). Marginal labels are shown in grey.

**Figure 5 pathogens-13-00961-f005:**
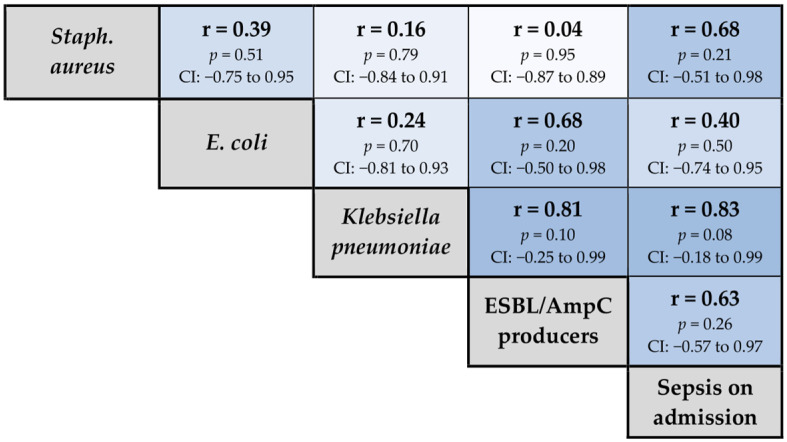
Correlation heatmap displaying pairwise Pearson correlation coefficients between individual blood culture isolates and the proportion of ICU patients with sepsis on admission. Colour intensity represents correlation strength and direction (darker shades correspond to stronger correlations, blue indicating positive correlation). Marginal labels are shown in grey.

**Table 1 pathogens-13-00961-t001:** Patient characteristics and unit activity.

Characteristic		2018/19 *	2019/20 *^#^	2020/21 *	2021/22 *	2022/23 *
Total admissions		1676	1651	1645	1614	1647
High-risk sepsis admissions from ward ^†^		12.7%	7.3%	4.5%	2.9%	8.1%
Unit-acquired infections in blood—per 1000 bed days ^‡^		1.8	2.3	6.4	5.3	3.3
Mean [SD] age—years		59.4 [18.4]	60.2 [18.1]	60.4 [16.6]	59.5 [17.3]	60.2 [17.2]
Male		988 [58.9%]	1006 [60.9%]	1032 [62.7%]	968 [60.0%]	947 [57.5%]
Mean [SD] APACHE II score		14.5 [6.3]	14.9 [6.5]	14.4 [5.6]	15.0 [6.3]	14.9 [6.1]
Admissions following trauma		206 [12.3%]	225 [13.6%]	148 [9.0%]	216 [13.4%]	204 [12.4%]
Mechanically ventilated ^¶^		853 [50.9%]	801 [48.5%]	732 [44.5%]	854 [52.9%]	824 [50.0%]
Sepsis ^§^		504 [30.1%]	441 [26.7%]	648 [39.4%]	511 [31.7%]	549 [33.3%]
Septic shock ^§^		114 [6.8%]	90 [5.5%]	97 [5.9%]	97 [6.0%]	120 [7.3%]
COVID-19 as primary or secondary reason for admission		0	18 [1.1%]	334 [20.3%]	156 [9.7%]	57 [3.5%]
Median [IQR] length of stay (unit survivors)—days		2.3 [1.1, 4.9]	2.4 [1.1, 4.7]	2.7 [1.3, 5.9]	2.5 [1.1, 5.5]	2.8 [1.5, 5.7]
Admission specialty	Clinical haematology	36	22	19	35	46
	Colorectal surgery	74	83	88	81	75
	Upper GI surgery	35	30	48	26	20
Mean [SD] bed occupancy		21.8 [2.0]	21.4 [2.4]	29.2 [12.9]	25.3 [3.2]	25.0 [3.1]
Risk-adjusted ICU mortality		0.78	0.85	0.89	0.88	0.79
Count of unique patients with bacteraemia detected on ICU		13	16	38	30	29

* April to March inclusive. ^#^ Appointment of an infection specialist. ^†^ Proportion of eligible admissions (unit admissions with infection from a ward in the same hospital) with four or more organ dysfunctions during the first 24 h following admission. ^‡^ Positive blood culture at least 48 h following unit admission in patients admitted to the unit for at least 48 h. ^¶^ During the first 24 h following admission. ^§^ According to the Sepsis-3 definitions and during the first 24 h following admission. Abbreviations: SD—standard deviation; IQR—interquartile range; GI—gastrointestinal.

**Table 2 pathogens-13-00961-t002:** Antibacterials in each WHO AWaRe classification group with the highest consumption (total DDDs per 1000 bed days).

Access	Watch	Reserve
Metronidazole (630.4)	Piperacillin with tazobactam (836.0)	Linezolid (90.2)
Co-amoxiclav (560.0)	Cefuroxime (511.4)	Daptomycin (39.0)
Co-trimoxazole (411.0)	Meropenem (480.1)	Tigecycline (19.1)
Flucloxacillin (221.2)	Vancomycin (342.6)	Ceftazidime with avibactam (2.6)
Doxycycline (202.7)	Azithromycin (197.3)	Ceftolozane with tazobactam (2.2)

## Data Availability

Reasonable requests for data sharing should be addressed to the corresponding author.

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
