# Peer review of "Exploring Antibacterial Usage and Pathogen Surveillance over Five Years in a Tertiary Referral Teaching Hospital Adult General Intensive Care Unit (ICU)"

_pathogens, 2024, doi:10.3390/pathogens13110961_

Round 1
Reviewer 1 Report
Comments and Suggestions for Authors
This manuscript thoroughly analyzes antibacterial usage and pathogen surveillance within a tertiary referral ICU over a critical five-year period, including the COVID-19 pandemic. It offers valuable insights into shifts in antimicrobial stewardship and resistance patterns within a teaching hospital's ICU. However, the following recommendations should be addressed:
- The manuscript's title is overly lengthy, which may confuse readers. I suggest shortening the title by removing "(including the COVID-19 pandemic)," which can be reflected in the text and keywords instead.
- It is recommended that additional studies be included in the introduction that specifically address pathogen surveillance in ICU settings during the COVID-19 pandemic.
- In the materials and methods section, the authors state, "As anonymized retrospective observational data were the focus of this project no ethics approval was required." Despite the plan to keep patient data anonymized, obtaining ethical approval is crucial to demonstrate a commitment to adhering to ethical standards, and it might be possible to secure this approval retrospectively.
- The authors should provide a clearer justification for the exclusion of certain antibiotics, such as erythromycin, from the analysis. The reason given, "it is predominantly prescribed locally (on a restricted basis)," does not sufficiently support the study's aim to explore antibacterial consumption partnered with pathogen surveillance.
- The authors need to expand their comparison of findings with global data on antimicrobial resistance trends. Including more references to international studies could emphasize the global relevance of your results within a One Health approach. I suggest the following two studies:
https://doi.org/10.1016/j.soh.2024.100077
https://doi.org/10.3390/microorganisms11010161
6. The conclusion is too brief and lacks specific outcomes from the study. The authors should elaborate more on the conclusions based on their findings.
Author Response
Dear Editors and Reviewer,
Thank you very much for taking the time to review our manuscript. Please find detailed responses below and the corresponding revisions/ corrections with changes tracked in the re-submitted manuscript.
|
This manuscript thoroughly analyzes antibacterial usage and pathogen surveillance within a tertiary referral ICU over a critical five-year period, including the COVID-19 pandemic. It offers valuable insights into shifts in antimicrobial stewardship and resistance patterns within a teaching hospital's ICU. However, the following recommendations should be addressed: The manuscript's title is overly lengthy, which may confuse readers. I suggest shortening the title by removing "(including the COVID-19 pandemic)," which can be reflected in the text and keywords instead. |
Amended as suggested. |
|
It is recommended that additional studies be included in the introduction that specifically address pathogen surveillance in ICU settings during the COVID-19 pandemic. |
Thank you for this suggestion – we have added some details of a large meta-analysis to support the aims of the study: Langford, B.J.; So, M.; Simeonova, M.; Leung, V.; Lo, J.; Kan, T.; Raybardhan, S.; Sapin, M.E.; Mponponsuo, K.; Farrell, F.; Leung, E.; Soucy, J-P. R.; Cassini, A.; MacFadden, D.; Daneman, N.; Bertagnolio, S. Antimicrobial resistance in patients with COVID-19: a systematic review and meta-analysis. The Lancet Microbe 2023, 4, e179-e191, doi: 10.1016/S2666-5247(22)00355-X. |
|
In the materials and methods section, the authors state, "As anonymized retrospective observational data were the focus of this project no ethics approval was required." Despite the plan to keep patient data anonymized, obtaining ethical approval is crucial to demonstrate a commitment to adhering to ethical standards, and it might be possible to secure this approval retrospectively. |
The sentence has been amended to read “As anonymised retrospective observational data were the focus of this project, an ethics waiver was approved by the local critical care governance lead.”. This reflects standard practice in UHS and indeed over the UK. We have also added confirmation to the text that the analysis was performed according to the Declaration of Helsinki which we hope demonstrates our commitment to adhering to appropriate ethical standards. |
|
The authors should provide a clearer justification for the exclusion of certain antibiotics, such as erythromycin, from the analysis. The reason given, "it is predominantly prescribed locally (on a restricted basis)," does not sufficiently support the study's aim to explore antibacterial consumption partnered with pathogen surveillance. |
Only a very small proportion of our patients (approximately 3.5%) were exposed to erythromycin and the average annual DDDs per 1,000 bed days was approximately 20. Because of challenges in administration and high risk of adverse effects; the sole indication for erythromycin in the ICU at UHS is for prokinetic purpose. This has been added with citation:- McKenzie, C.A.; Barton, G.; Philips, B.J. (eds). MedicinesComplete — CONTENT > Critical Illness > Drug: Erythromycin; London: Pharmaceutical Press, 2022.Available online: https://www.medicinescomplete.com/#/content/critical/451446021 |
|
The authors need to expand their comparison of findings with global data on antimicrobial resistance trends. Including more references to international studies could emphasize the global relevance of your results within a One Health approach. I suggest the following two studies: https://doi.org/10.1016/j.soh.2024.100077
https://doi.org/10.3390/microorganisms11010161 |
Thank you for this suggestion and for including relevant references, this is most helpful. We have included these, highlighting the importance of a “OneHealth” approach and linking this to our work at UHS. |
|
The conclusion is too brief and lacks specific outcomes from the study. The authors should elaborate more on the conclusions based on their findings. |
Thank you for the suggestion. We have added more detail from the manuscript for the key outcomes of interest, namely proportion of “access” category antibacterial usage, most concerning antimicrobial resistance and mortality. Although, with respect we believe that a short overal conclusion fits the purpose of our manuscript |
Reviewer 2 Report
Comments and Suggestions for Authors
Line 29-32 mentions “However, the prevalence of most concerning antimicrobial resistance, with respect to pathogens, and mortality rate remained satisfyingly consistent suggesting the positive consequences of real-world antibiotic stewardship in an intensive care setting.”
Questions: how was prevalence measured, I mean what was the test, or data to claim prevalence remains same, was it measured as percentage (better to plot it if you can) ? Have you also looked for correlation of pathogen reporting and antimicrobial resistance (AMR) with MORTALITY ? I don’t see mortality data even anywhere in manuscript, what was mortality over the course of these 5 years. Can you also plot mortality verses all risk factors like sepsis, AMR, isolates.
Was there correlation between Sepsis and any of the isolate reporting or any isolate reporting ?
What was percentage mortality over the course of these five years ? I think getting correlation plot for all the parameters can address these questions and of there is any strong / significant correlation coming, should be reflected in the abstract of the manuscript.
Line 267-273: please clarify here, was that mortality reported in COVID patients or all admitted patients.
Line 68-69 mentions “As anonymized retrospective observational data were the focus of this project no ethics approval was required.”
Question is, was this study submitted to ethics/ medical review board office and they recommended about no need of such approvals ? If not, may double-check with the office once.
Table 1 shows no covid reported until year 2021. But what was the percentage of COVID 19 cases?
It will be useful to get correlation analysis for the study by splitting data into two groups (Those who reported COVID and those who were not reported with COVID). It may give better insights as how did COVID 19 has impacted isolates, AMR if any, Mortality etc.
Even table 1 presents different abbreviations for study years 2019/ 2020 which is not clear idea for what that means. Does it mean time from January 2019 – January 2020 ? or April 2019- April 2020. Even some figures show the similar confusing abbreviations. Please clearly describe these in the captions, as what they exactly mean. And I suggest to keep the span period for analysis (figures and tables) same if it is not.
Line 157: of admissions of admissions. Please correct it.
Line 161-162 claim that “During the peak of the COVID-19 pandemic (2020/21), mean maximum daily unit bed occupancy increased from 21.67 ±2.2 to 29.2 ±12.9 patients with a maximum bed occupancy of 67 patients (22/1/2021; Figure 1).”
It’s difficult to understand, how does bed occupancy increased in 2020/2021 when total admissions are showing-up to be second most less (1645) which was higher in several other years which are labeled as non-peak years. I think lines / or data need refinement. What does bed occupancy means compared to admission rates? What is the reasoning for this prolonged bed occupancy ? So in this context, please plot percentage of COVID patients too in figure 3. On the same time it is also important to comment what were the patient presentations/ diseases who had longer bed times? Were they COVID ? table 1 shows they were not COVID. Please clearly state here that they were not covid so as to avoid confusion for the readers that long bed times are not because of having covid. Please reflect the same changes in abstract too. They were not COVID then what was the reason that they took longer time on beds ? Please comment on all these things here.
What was the reason for admission ? Is it possible to reflect this information in table 1 ? It will be useful because may be patients in COVID peak were diagnosed with COVID.
Line 247: Blood cultures increased by 77% over the five years. The line is really wage, can you be more specific and detailed here. Do you mean that samples sent for testing/ culture increased to 77% or infection rate increased to 77% if you mean that only samples sent to testing facility increased ? what was the reason for that increase, please comment on it. Or doctors just became more vigilant.
Author Response
Dear Editors and Reviewer,
Thank you very much for taking the time to review our manuscript. Please find detailed responses below and the corresponding revisions/ corrections with changes tracked in the re-submitted manuscript.
|
Line 29-32 mentions “However, the prevalence of most concerning antimicrobial resistance, with respect to pathogens, and mortality rate remained satisfyingly consistent suggesting the positive consequences of real-world antibiotic stewardship in an intensive care setting.” Questions: how was prevalence measured, I mean what was the test, or data to claim prevalence remains same, was it measured as percentage (better to plot it if you can) ? Have you also looked for correlation of pathogen reporting and antimicrobial resistance (AMR) with MORTALITY ? I don’t see mortality data even anywhere in manuscript, what was mortality over the course of these 5 years. Can you also plot mortality verses all risk factors like sepsis, AMR, isolates. |
These are really valid points- thank you. With regard prevalence, we agree and have amended this from “prevalence” to “incidence” throughout (in the abstract, discussion and conclusion) for clarity. Thank you, this again is very important point, the mortality data (risk adjusted) was extracted from national (UK) audit reports (Intensive Care National Audit and Research Centre, ICNARC). We have now included in the results section (Table 1) and in discussion. This is a good suggestion below is a draft version of your suggestion. We believe that this does not add substantially to the manuscript. |
|
Was there correlation between Sepsis and any of the isolate reporting or any isolate reporting ? |
The graph below shows no clear correlation with a single isolate and sepsis.
|
|
What was percentage mortality over the course of these five years ? I think getting correlation plot for all the parameters can address these questions and of there is any strong / significant correlation coming, should be reflected in the abstract of the manuscript. |
Please see first response above. Added to results and discussion of manuscript.
|
|
Line 267-273: please clarify here, was that mortality reported in COVID patients or all admitted patients. |
Thank you. The proportions reported in the manuscript are for all patients – we have added the word “overall” to clarify this. Mortality related to SARS CoV2 admissions in this ICU was outside of the scope of this article but we are pleased to refer the reviewer to an earlier manuscript from our group. (https://doi.org/10.1002/wsbm.1577). |
|
Line 68-69 mentions “As anonymized retrospective observational data were the focus of this project no ethics approval was required.” Question is, was this study submitted to ethics/ medical review board office and they recommended about no need of such approvals ? If not, may double-check with the office once. |
For clarity, the sentence has been amended to read “As anonymised retrospective observational data were the focus of this project, an ethics waiver was approved by the local critical care governance lead.” The local governance lead is Dr. Thomas Price. This reflects standard practice in the University Hospital Southampton NHS Foundation Trust and indeed in UK.
|
|
Table 1 shows no covid reported until year 2021. But what was the percentage of COVID 19 cases? |
The percentage of ICU admissions with SARS CoV2 as primary or secondary reason for admission is described in Table 1: 9.7% in 2021/22 and 3.5% for 2022/23. The percentage of COVID-19 figures were not reported in the 2020/21 ICNARC case mix programme report for UHS; therefore we did not include in Table 1 and hence the denotation “NR” for 2020/21 in the body of the table. |
|
It will be useful to get correlation analysis for the study by splitting data into two groups (Those who reported COVID and those who were not reported with COVID). It may give better insights as how did COVID 19 has impacted isolates, AMR if any, Mortality etc. |
Thank you, we agree that this could be of potential interest but believe that this has been previously described in the unit’s COVID-19 response/ outcomes article cited above (https://doi.org/10.1002/wsbm.1577). |
|
Even table 1 presents different abbreviations for study years 2019/ 2020 which is not clear idea for what that means. Does it mean time from January 2019 – January 2020 ? or April 2019- April 2020. Even some figures show the similar confusing abbreviations. Please clearly describe these in the captions, as what they exactly mean. And I suggest to keep the span period for analysis (figures and tables) same if it is not. |
Thank you very much – this is a valid point and we apologise. There is inconsistency in the original submitted manuscript. The correct timespan is April to March inclusive for the entire five-year period. The first footnote of Table 1 clarifies this for this data. A similar note has been added to the legends for Figures 2 and 3. |
|
Line 157: of admissions of admissions. Please correct it. |
Corrected – thank you. |
|
Line 161-162 claim that “During the peak of the COVID-19 pandemic (2020/21), mean maximum daily unit bed occupancy increased from 21.67 ±2.2 to 29.2 ±12.9 patients with a maximum bed occupancy of 67 patients (22/1/2021; Figure 1).” It’s difficult to understand, how does bed occupancy increased in 2020/2021 when total admissions are showing-up to be second most less (1645) which was higher in several other years which are labeled as non-peak years. I think lines / or data need refinement. What does bed occupancy means compared to admission rates? What is the reasoning for this prolonged bed occupancy ? So in this context, please plot percentage of COVID patients too in figure 3. On the same time it is also important to comment what were the patient presentations/ diseases who had longer bed times? Were they COVID ? table 1 shows they were not COVID. Please clearly state here that they were not covid so as to avoid confusion for the readers that long bed times are not because of having covid. Please reflect the same changes in abstract too. They were not COVID then what was the reason that they took longer time on beds ? Please comment on all these things here. What was the reason for admission ? Is it possible to reflect this information in table 1 ? It will be useful because may be patients in COVID peak were diagnosed with COVID. |
Thank you. As the reviewer notes; we have included in our text that admission numbers were fairly consistent over the entire five years. We agree with the reviewer that ICU length of stay has increased and thus bed occupancy. We have not investigated the cause of the increased length of stay or whether this is related to COVID-19 ICU admissions. We do not believe this information is vital at this time but agree that COVID-19 could be a potential explanation for the increased ICU length of stay. Although there are other factors related to ICU admission types we receive, including that we receive a high proportion of heamatoncology and complex trauma admissions. In the discussion, we have added detail describing the proportion of unit beds that were assigned to patients with COVID-19 admissions. This we hope demonstrates that much of the ICU work during this period was associated with COVID-19 ICU admissions: “Prior to this, the proportion of commissioned beds occupied by a patient was approximately 90%. During the peak of the pandemic the number of available beds on the ICU in-creased to 76; of these, 55 were exclusively for treating patients with COVID-19 infection. The consequent decrease in staff-to-patient ratio, expansion of the unit to external areas of the hospital and sharing of bed spaces may have impacted multiple factors, including antimicrobial resistance.” |
|
Line 247: Blood cultures increased by 77% over the five years. The line is really wage, can you be more specific and detailed here. Do you mean that samples sent for testing/ culture increased to 77% or infection rate increased to 77% if you mean that only samples sent to testing facility increased ? what was the reason for that increase, please comment on it. Or doctors just became more vigilant. |
Thank you, we believe that the statement in the manuscript (“the annual number of blood cultures analysed increased by 77% over the five-year period”) is clear that this refers to the number of samples sent for testing and not the infection rate. We have added a comment about the potential reasons for this increase to the manuscript, including delayed presentation because of COVID-19, an increased awareness of sepsis and more complex surgeries (associated with delayed presentation). |

Reviewer 3 Report
Comments and Suggestions for Authors
antibacterial consumption, partnered with pathogen surveillance, over a five-year period (2018 to 2023) in a tertiary referral adult general intensive care unit (ICU).
Concise well written introduction wit clear aim.
Is SDD used in your ICU’s?
It appears as though you looked at all admissions – correct? This appears to be reflected in your median LOS 2.3-2.8 days & APACHE 11 score [Table 1]. The admissions of those with LOS<24 hours would represent a low risk ‘observation group’. The LOS becomes relevant as most of the duration of antibiotic use that you mention at lines 195-208 exceed the median ICU LOS.
Presumably an adult ICU – no paediatric admissions?
For table 1 – the proportion of those receiving Mechanical ventilation [a high risk group] or those with central lines – would be of interest if you have that [Hurley JC. Trends in ICU mortality and underlying risk over three decades among mechanically ventilated patients. A group level analysis of cohorts from infection prevention studies. Annals of Intensive Care. 2023 Jul 11;13(1):62].
In table 1 the mean be occupancy is unclear – I presume this is a occupied bed count – what is the maximum? Is it really 67? That mean you only have approx. 30% occupancy or was thre a change in bed numbers with the pandemic?
Unfortunately the changes have not been tested for statistical significance and visually there appears to be little change over time – Presumably there was no change in the antimicrobial stewardship practices over this time period – correct? Probably a more relevant question is how much deviation from guidelines was there?
The discussion in line 267 – 277 is speculative and possibly beyond what you have evidence for.
Author Response
Dear Editors and Reviewer,
Thank you very much for taking the time to review our manuscript. Please find detailed responses below and the corresponding revisions/ corrections with changes tracked in the re-submitted manuscript.
|
antibacterial consumption, partnered with pathogen surveillance, over a five-year period (2018 to 2023) in a tertiary referral adult general intensive care unit (ICU). Concise well written introduction wit clear aim. Is SDD used in your ICU’s? |
This is an important question. We confirm selective digestive decontamination (SDD) has never been used in ICU at UHS.
|
|
It appears as though you looked at all admissions – correct? This appears to be reflected in your median LOS 2.3-2.8 days & APACHE 11 score [Table 1]. The admissions of those with LOS<24 hours would represent a low risk ‘observation group’. The LOS becomes relevant as most of the duration of antibiotic use that you mention at lines 195-208 exceed the median ICU LOS. |
Yes, we confirm that we included all ICU admissions to our general intensive care unit (GICU). Although we did not included admission to our specialist cardiac intensive care unit (CICU) and neurosciences intensive care unit (NICU). We agree that short ICU admissions or antibacterial courses may have influenced our data but note that only small proportion of antibacterial courses sampled were <=1 day (e.g. 28% of all antibacterial courses). We therefore decided to include all courses for completeness and believe that the addition of the mechanical ventilation data (see response below; approximately 50% of admissions mechanically ventilated) helps to further define our patient cohort. |
|
Presumably an adult ICU – no paediatric admissions? |
The reviewer is correct that this is an adult ICU as per the last sentence of the introduction: “in a tertiary referral adult general intensive care unit (ICU)”. The hospital has a separate paediatric ICU and their data not included is not included in this manuscript. |
|
For table 1 – the proportion of those receiving Mechanical ventilation [a high risk group] or those with central lines – would be of interest if you have that [Hurley JC. Trends in ICU mortality and underlying risk over three decades among mechanically ventilated patients. A group level analysis of cohorts from infection prevention studies. Annals of Intensive Care. 2023 Jul 11;13(1):62]. |
Thank you – the proportion of admissions mechanically ventilated during the first 24 hours following admission (consistent throughout period of the evaluation) has been added to Table 1. |
|
In table 1 the mean be occupancy is unclear – I presume this is a occupied bed count – what is the maximum? Is it really 67? That mean you only have approx. 30% occupancy or was thre a change in bed numbers with the pandemic? |
Thank you, we agree; this is mean daily bed occupancy rate count. This has been further clarified in the discussion too, namely that the number of beds increased temporarily during the pandemic. |
|
Unfortunately the changes have not been tested for statistical significance and visually there appears to be little change over time – Presumably there was no change in the antimicrobial stewardship practices over this time period – correct? Probably a more relevant question is how much deviation from guidelines was there? |
Our analysis is purely descriptive and believe that inferential statistics would be inappropriate for a single centre evaluation such as this. We also believe that the evidence that we do report minimal change over time is testament to our antimicrobial stewardship and strong multidisciplinary team approach. The unit has expanded by 6 beds in April 2022 (see text). Since the end of 2019, a single permanent microbiologist, with responsibility for antimicrobial stewardship and infection management in ICU, has been in post (this has been added to Table 1). At this point, regular antimicrobial stewardship and infection MDTs were introduced. Therefore, based on local audit results, deviation from guidelines and expert advice is known to be minimal. We also agree that deviation from guidelines is a relevant question and this will be focus of our next collaborative project. We have added this sentence to discussion: “Looking forward our future work will include guideline adherence for specific indication and medicines optimisation.”. |
|
The discussion in line 267 – 277 is speculative and possibly beyond what you have evidence for. |
We agree that some of this is speculative and have carefully selected our language to identify the parts that are purely speculation. The first sentence (starting “Additionally, the mortality rate…”) is factual. The second sentence is clearly labelled as speculation: “This is possibly due to…”. The reference in the following sentence supports the statement that there was potential overuse of antibiotics in patients with COVID-19 infection and we think it is reasonable to speculate (appropriately marked: “… may have contributed…”) that this may increase AMR which is associated with increased mortality. The final sentence is a description of ICU clinical practice and our considered reflection of this. |
Round 2
Reviewer 2 Report
Comments and Suggestions for Authors
COMMENT 1: I can see there could be useful information for correlation of sepsis with isolates. Is that table a normalized data or average of what was there for each year in the ICU ?
The graph below shows no clear correlation with a single isolate and sepsis.
COMMENT 2:
How was no clear correlation determined ? Was there any statistical test or correlation matrix used in this case to reach that conclusion. I suggest run multivariate correlation test/ analysis. This kind of matrix is an example: https://images.app.goo.gl/BWgTuG69Y59Z4sx5A .
If interested, it will get you some useful information out of these two plots which could be a secondary useful outcome of the study in addition to current objective.
COMMENT 3:
Not Reported in medical language means no COVID reported in those 1000s of patients admitted in ICUs each year even when it was COVID peak time 2020. ICNARC report was missing COVID-19 figures in 2020/2021, can you better define these words in manuscript rather than simply NR.
From above reply it feels like, ICNARC report was missing COVID-2019 data from those years or data not available, whereas writing it NR means COVID 19 was tested and ruled out but was not reported in those patients. Could there be any possible reason that the report has missed that data ? Whatever, the case, NR seems to be an inappropriate terminology for defining the current situation of COVID incidence in those years. Or the term NR needs to be defined in more details. I understand your focus was not COVID in this study but it is conveying some different meanings.
There are some additional comments below. I hope they can improve the manuscript.
Line 110: form
Lines 267-270: You infected cases are comparatively extremely low and even in general the total infected patients was extremely low to reach such conclusions. Better if you can comment the cons of your study here compared to the study in contrast (reference 13)
The line 60-61: The impact of the COVID-19 pandemic on antibacterial consumption, drug resistance and length of stay has not been fully described.
Can you add here some literature about these topics as what is known and unknown thereby making a link as how this study is helping address the gap? Also it will be great to include a definition of pathogen surveillance and antimicrobial stewardship in the beginning of introduction.
I googled that line and there were several studies already which appeared, some specific are listed here but I suggest to get a deep dive into current literature and access evaluate pros and cons of your study in comparison to the existing literature. I am concerned about the aim of this study due to really low number of patients who were evaluated for infection incidence and then the number of isolates too are less to reach conclusions as what was the impact of COVID-19.
Table 1: may include how many patients were detected with infections in each year.
@font-face {font-family:"Cambria Math"; panose-1:2 4 5 3 5 4 6 3 2 4; mso-font-charset:0; mso-generic-font-family:roman; mso-font-pitch:variable; mso-font-signature:-536870145 1107305727 0 0 415 0;}p.MsoNormal, li.MsoNormal, div.MsoNormal {mso-style-unhide:no; mso-style-qformat:yes; mso-style-parent:""; margin:0in; mso-pagination:none; text-autospace:none; font-size:11.0pt; font-family:"Arial",sans-serif; mso-fareast-font-family:Arial;}.MsoChpDefault {mso-style-type:export-only; mso-default-props:yes; font-size:11.0pt; mso-ansi-font-size:11.0pt; mso-bidi-font-size:11.0pt; font-family:"Calibri",sans-serif; mso-ascii-font-family:Calibri; mso-ascii-theme-font:minor-latin; mso-fareast-font-family:Calibri; mso-fareast-theme-font:minor-latin; mso-hansi-font-family:Calibri; mso-hansi-theme-font:minor-latin; mso-bidi-font-family:"Times New Roman"; mso-bidi-theme-font:minor-bidi; mso-font-kerning:0pt; mso-ligatures:none;}.MsoPapDefault {mso-style-type:export-only; mso-pagination:none; text-autospace:none;}div.WordSection1 {page:WordSection1;}

Round 3
Reviewer 2 Report
Comments and Suggestions for Authors
Which test was used ? Pearson, or what ? I found some important insights in the plots below.
Mortality has 0.7 correlation with incidence of isolates from blood cultures. Not only that, but the reason for mean bed occupancy is sepsis (0.96), can you do it by adding the COVID data here. And discuss the results if covid has any impact.
Because the abstract mentions these lines: The COVID-19 pandemic peak year was associated with increased ICU bed occupancy, a greater number of positive blood cultures but lower antibacterial consumption. If you add covid data into here and run a correlation analysis, you probably would be able to get some reasoning (as what are the factors showing correlation with patient’s long stays especially if COVID is the reason or association of COVID with sepsis) with a better statistics rather writing it in a wage manner. I suggest you to add another column which represent how many patients co-presented COVID+ sepsis, then run correlation test, it will be able to get you better idea and confidence for the claims and statements which are really baseless for now.
The statement sounds like, COVID is the reason for long stays. Can you better refine it, since no one has any idea for that long stay ? May be they had sepsis that made them to stay longer ? But without running a correlation test, it is difficult to say.
Pearson coorelation shows several other insights. Correlation with Klebsiella. Incidence for coexistence of Klebsiella and S. pneu has high correlation.
I encourage you if you can make some professional plots and discuss the new results.
Resistance is reported having correlation with Klebsiella, E. coli, S. pneu (strong correlations 0.68-0.81). Interestingly, the resistance could also be reason for sepsis (correlation 0.63, moderate correlation).
Where you have discussed the results from other studies about these characteristics., these results can also be discussed at the same place. Those having high correlation can be pointed in abstract.
